# ML-Based Delay Attack Detection and Isolation for Fault-Tolerant Software-Defined Industrial Networks

**DOI:** 10.3390/s22186958

**Published:** 2022-09-14

**Authors:** Sagar Ramani, Rutvij H. Jhaveri

**Affiliations:** 1Department of Computer Engineering, Gujarat Technological University, Ahmedabad 382424, India; 2Department of Computer Science & Engineering, Pandit Deendayal Energy University, Gandhinagar 382007, India

**Keywords:** SDN, delay attack, security, machine learning, industrial networks, CPS

## Abstract

Traditional security mechanisms find difficulties in dealing with intelligent assaults in cyber-physical systems (CPSs) despite modern information and communication technologies. Furthermore, resource consumption in software-defined networks (SDNs) in industrial organizations is usually on a larger scale, and the present routing algorithms fail to address this issue. In this paper, we present a real-time delay attack detection and isolation scheme for fault-tolerant software-defined industrial networks. The primary goal of the delay attack is to lower the resilience of our previously proposed scheme, SDN-resilience manager (SDN-RM). The attacker compromises the OpenFlow switch and launches an attack by delaying the link layer discovery protocol (LLDP) packets. As a result, the performance of SDN-RM is degraded and the success rate decreases significantly. In this work, we developed a machine learning (ML)-based attack detection and isolation mechanism, which extends our previous work, SDN-RM. Predicting and labeling malicious switches in an SDN-enabled network is a challenge that can be successfully addressed by integrating ML with network resilience solutions. Therefore, we propose a delay-based attack detection and isolation scheme (DA-DIS), which avoids malicious switches from entering the routes by combining an ML mechanism along with a route-handoff mechanism. DA-DIS increases network resilience by increasing success rate and network throughput.

## 1. Introduction

Distinct types of Internet of Things (IoT) devices have been deployed in recent years, and the linked interactions between them have become more diverse and intricate, posing new challenges in IoT device communication and device management. In addition to this industrial cyber-physical system (ICPS) is an intelligent pneumatic environment, formed by an amalgamation of computing networking, and physical dimensions. ICPS demands that new standards are linked to flexibility, self-optimization, self-recoverability, and heterogeneity without sacrificing the quality of service (QoS) [1]. Software-defined networks (SDNs) can be a great asset in managing the communication between IoT and ICPS devices and addressing the challenges [2,3]. By exploiting software or hardware flaws, an attacker can compromise a network forwarding device and affect the network resilience [4]. An attacker with access to compromised forwarding devices may be able to take complete control of an SDN-enabled network [5]. Network resilience is vital for time-critical applications to provide fault tolerance [6], especially at the control plane, during real-time communication in the network [7]. As malicious switches may alter the packets, the data plane invasion operates as a threat causing the control plane’s misbehavior [8].

Scientists from Threatcare and IBM found security loopholes in smart city gadgets, which are used everywhere from traffic analysis to radiation detection [9]. Additionally, these attacks can easily proliferate over IoT networks. Therefore, isolating various types of communication or segmenting networks is a useful approach to limiting the lateral transmission or expansion of attacks [10,11]. In SDN networks, it is critical to isolate malicious switches in real time since the entire SDN network is highly vulnerable due to the centralized control plane [12]. In addition to this, meeting the delay requirements of network flows while effectively offering fault resilience against smart intrusion is a major challenge in time-critical applications [13]. Thus, there is a need to maintain network resilience and security by not only identifying the threats but also isolating these threats by applying a reactive strategy in real time. Therefore, in this paper, we consider an attack in which the adversary compromises an OpenFlow switch to maliciously delay the LLDP packets [8].

Industrial networks demand high resilience to achieve a high quality of service (QoS). Therefore, there is a need to detect any unusual activity in real-time which may degrade the overall network performance and can cause disaster. OpenFlow has added a statistic checking tool that may be used to spot unusual activity and pinpoint a rogue switch on the SDN data layer to solve this problem [14]. It is challenging to synchronize independent OpenFlow counters from various switches because the OpenFlow specification does not support a specific upper limit inside which a switch must execute the stat-req intimations from the control plane [15]. Therefore, it is difficult to isolate any OpenFlow switch from the network in real time. In the context of this problem, we devised a scheme for detecting malicious switches at the control plane using ML classifiers which can aid the SDN controller in making reactive decisions. The ML-based scheme not only detects but also isolates the malicious switches in the route which intentionally delay LLDP packets.

Several studies [16,17,18,19] propose exploring the configurability of SDN switches in real-time to enhance the overall QoS of SDN. Maintaining network resilience in the face of unintentional failures and malicious attacks is a significant factor for critical infrastructures such as power grids and industrial CPS [20]. To address resilience in the industrial CPS, we previously presented delay [4] and bandwidth-based [21] resilience mechanisms for enhancing the fault-tolerance capabilities which use contract-based mechanisms to address demands of critical applications and to provide zero downtime. However, these approaches do not consider the all-important security aspect. Numerous methods have been proposed to secure the data plane, which include (1) acknowledgment-based mechanisms, (2) packet probing, (3) flow statistics, and (4) cryptographic mechanisms [22]. However, SPHINX [23] and WedgeTail [24] are some examples of frameworks for identifying and hunting forwarding devices that fail to process packets as expected in the data plane. The majority of these solutions cause significant communication overhead because they are primarily concerned with anomaly detection at the first and last hops of a network. As security is one of the key aspects of resilience, in this work, we consider addressing delay-based attacks, which may cause disaster in time-critical industrial and healthcare applications [25]. Therefore, we extend our work proposed in [4,8] to isolate malicious switches, which are compromised with the delay-based attack. However, we confront the following research challenges from the literature:*Challenge 1: * How will the QoS parameters (success rate and average network throughput) behave under the existing approach in the presence of network fault events (link break event and dynamic changes in delay requirements of the contract event) in the presence of intrusion and congestion?

*Challenge 2:* Eventually, if the scale of the attack increases, which type of impact will it have on the QoS parameters (success rate and average network throughput) in the presence of network fault events (link break event and dynamic changes in delay requirements of the contract event)?

The existing works do not address delay-based attacks for time-critical applications. Therefore, we address these security and fault-resilience challenges with the following contributions:We represent a KNN-based detection and isolation mechanism to counter the delay-based attack on an OpenFlow switch.The route-handoff mechanism is employed to isolate the malicious switches, which are part of the current transmission route.We compare and analyze the performance of the proposed scheme with our previous approach, SDN-RM, under normal and attack conditions.The experiment result shows that delay attack could not influence the average network throughput and success rate with compromised switches. Moreover, the existing approach provides an efficient solution under specified conditions.

The remainder of the paper is structured as follows. Section 2 examines relevant recent works. The delay-based attack detection and isolation system (DA-DIS) framework is presented in Section 3. Comparative results analysis of the DA-DIS and SDN-RM approaches is presented in Section 4. The entire study is concluded in Section 5.

## 2. Related Work

In recent years, SDN security literature has mostly focused on the positioning of the SDN controllers and security applications along with real-time verification of network limitations. However, none of them address security issues for compromised forwarding devices.

### 2.1. Review of SDN Attack Defence Mechanisms

In [26], the authors present an approach where an SDN controller detects corrupted switches using real-time information analysis. Packet switching and packet dropping are two types of harmful behaviors investigated in the study. However, the consequences of malicious switches, which lead to false information in the statistics reports, are not presented. SPHINX is one of the options for protecting the SDN data plane that does not rely on the trustworthiness of switches. SPHINX’s key drawback is that it is unable to determine whether packet delays are caused by malicious switches. Additionally, to identify attacks, the detection process mostly relies on policies defined by an administrator. Another security method is active probing, which employs extra test flows to identify or locate a malicious switch. With the expansion of the network, more test flows and rules are put into each switch’s ternary content-addressable memory (TCAM). The greater the number of test flows, the greater the overhead and the greater the quantity of TCAM, which is undesirable because TCAM is expensive and power-demanding [27]. The capture of malicious switches that do not adhere to the OpenFlow protocol or malicious hosts that do not adhere to the ARP protocol are presented in the BEADS framework [28]. BEADS does not identify the full stealthy attacks that are ready to attack a target. Instead, it discovers tactics that have a substantial influence on the network as a result of one or more problems, similar to stack-overflow vulnerabilities. However, writing an exploit that leverages the fault in a focused manner still requires manual effort. In [29], the authors represented a novel approach to detecting the warm-whole attack in wireless sensor networking. The main purpose of the attack is to convince the controller that two vulnerable non-neighboring switches are physically coupled. This allows the controller to manage various traffic flows, which initially should not travel through the compromised switches as the forged link leads to shorter ideal paths. The attacker might use this exploit as a springboard to conduct other DoS attacks or steal sensitive information. To detect the attack the approach uses the delay time between two switches. This connection is regarded as a false link if the proportion of timeout flows is higher than the predetermined threshold, it isolates the false links from the network.

Some techniques for detecting and mitigating harmful devices within a network either presume a basic threat model [30] or incur significant expenses even during normal operations [31]. When faced with advanced threats, the ATPG network diagnostic tool [30] may incorrectly attribute harmless entities since it usually assumes a superficial situation in which errors appear regularly. On the other hand, while cryptographic-based verification protocols, such as OPT [31], can guarantee route approval in the face of powerful attackers, their high overhead may make them unsuitable for widespread implementation. Machine learning-based detection strategies for malicious switches are few and most are detection techniques for malicious switches. Zhou et al. present CRAD (crowd-aware approach) [32], for detecting rogue access points in disguise without requiring additional hardware. CRAD employs RSS spatial correlation to identify the probable masquerader who should be in a distinct position from the real ones. RSS measurements gathered from the general public aid in the construction of a strong profile and the reduction of the inaccurate influence of individual RSS results. The findings reveal that CRAD can dynamically match the contour lines to filter out the identified aberrant samples in real-time. To identify the malicious switches, Kuo et al. [33] presented a better and more practical method for detecting malicious switches. The model’s applicability is demonstrated in the client, making deployment easier, with higher security, and a higher detection rate. Huang et al. [34] provide an extreme learning machine-based classifier along with a computationally effective method for identifying traffic signs. Zhong et al. [35] proposed a rapid Gaussian kernel learning approach that can converge a universal result for any classification job. Gore et al. [36] suggested a strategy that involves applying the Markov Chain method to obtain descriptions of detected cyber threats to make intelligent defensive actions that maximize the utilization of scarce resources. The key finding of this approach is to discover previously undiscovered themes of widespread vulnerability.

These existing works on SDN security do not focus on securing the network from a delay-based attack launched by compromising switches [37,38,39,40].

### 2.2. Review of SDN-Based Fault-Tolerance Mechanisms

Babiceanu et al. [41] suggested a holistic modeling environment, which addresses the assurance of virtual industrial systems via resilience methods and security. The study looks at the ability to scale ontology to categorize and detect system risks, vulnerabilities, and threats. Resilience is inextricably linked to ideas such as danger, risk, and vulnerability. As a result, current research intends to provide a foundation for developing methods for IIoT systems to attain equilibrium resilience in the context of cyber security assaults. The design of FT-SDN [42] comprises a straightforward and efficient decentralized control plane with many controllers. The internal states of the controller are regularly updated by FT-SDN via a coordinated process. In the event of a breakdown, FT-SDN can choose an alternative operational controller depending on the latency and distance between various network elements. In [43], the authors explored the distributed control architecture of the SDN’s fault-tolerant arbitration challenge. This guarantees the accuracy of the computation outcomes regardless of whether any controller is faulty or attacked. In addition, several attack vectors that might allow the exploitation of SDN vulnerabilities are described in [44] to support the argument that safe and reliable SDNs should be built by design.

## 3. Delay Attack Detection and Isolation System (DA-DIS)

In this section, we propose a delay attack detection and isolation scheme (DA-DIS).

### 3.1. Overview of SDN-RM

As mentioned previously, our existing work is an extension of our previous work, which is SDN-RM [4]. SDN-RM is a contract-based technique that intends to fulfill the end-to-end timing constraints of all the communications flow in SDN systems. The architectural design of SDN-RM comprises four components that aid to create resilience in the SDN.

#### 3.1.1. Contracts and Observers

In this paradigm, a contract explicitly describes (1) hypotheses on the inputs and environment, (2) inputs and outputs of an element, (3) assurances on the outputs of an element, and (4) factors that might be used for run-time modifications in contracts. We consider two types of contracts: strong contracts and weak contracts. A strong contract ensures the end-to-end delay guarantee from a source si to a destination sj by considering the strong parameter values, while a weak contract is defined in a similar way with a weaker set of parameter values.

The observer is responsible for a certain contract to confirm that the component produces the desired behavior. A fault is notified in the event that a contract failure is discovered by the corresponding observer. An observer is unaffected by the behavior of the component.

#### 3.1.2. Monitors

Monitors are responsible for the inspection of two events: (1) the link break event (E1) and (2) dynamic changes in delay requirements of the contract event (E2). Using the switch port data that were obtained, a link break monitor in the SDN controller may identify a link break event (E1). By tracking contract changes, the monitor identifies a contract modification (E2) in the run-time delay needs.

#### 3.1.3. Resilience Manager

The resilience manager’s (RM) goal is to make the network more fault-tolerant. It has control logic that processes the desired reaction plan after receiving the reported fault event. We consider three types of reaction plans when the fault is reported in the network.

RP1—path reassignment and path recalculation: In order to redirect the network flows, the path-finding algorithm is used to recalculate the path between the required source–destination pair. The SDN controller applies the updated forwarding rules for the alternate pathways to the appropriate switches.RP2—transition to a weak contract: If the delay criteria of the strong contract cannot be guaranteed after route recalculation; the RM can choose a transition to a weak contract.RP3—sending out a warning: In the worst-case scenario, a warning is sent when the weak contract’s delay conditions cannot be met (weak contract failure).

#### 3.1.4. Path-Finding Algorithm

The route finding algorithm is in charge of determining the path having the least amount of end-to-end delay. It is triggered (1) whenever a new flow enters the system, (2) regularly to propose a new route depending on the condition of the network, and (3) if an observer reports a failure. The newly proposed route is subsequently used by the SDN controller for the current flows. To determine the route with the smallest delay between the origin and the endpoint, the path-finding method uses Dijkstra’s shortest path algorithm.

### 3.2. Operations of DA-DIS

Our framework is built on the CPS communication network, comprising SDN network architecture that programs OpenFlow switches using open flow protocols. States and flows can change at any moment in CPS networks, and requirements may also change depending on the network state. As a result, with this sort of system, where the system’s environment changes on a regular basis, security is a vital issue. Furthermore, in real-time, it is difficult to discover and remediate systems that are under threat. Such threats have a direct impact on the QoS parameter, resulting in a decrease in system performance and robustness. On the controller’s side, the controller has a complete topology view and may obtain any statistics. The suggested method aims to identify the malicious network switches that intentionally delay LLDP packets. If a switch is discovered to be malicious, the suggested technique will isolate it from the routing path and find an alternate route with the route-handoff mechanism. As shown in Algorithm 1, Contract_Observer is responsible to notify the fault to the monitor from the topology. While Check_Status_Change is responsible to notify the status of the switch from the topology. In addition to this, it is responsible for detecting the Delay_attack. By doing so, it can obtain the model decision based on whether to isolate the switch or not from the routing path by calling k_shortest_path. However, while calling k_shortest_path it makes sure that a malicious switch is not a part of the routing path.
**Algorithm 1** DA-DIS detection and isolation algorithm**Input**: Current Topology G = (Sw,E)**Output**: Path/Backup path (P ${}_{i}$) between source-destination node pairContract_Observer(Sw ${}_{i}$, Port P ${}_{i}$)Status Sw ${}_{m}$  = Check_Status_Change();for all Sw ${}_{i}$  ∈ G do   if Check_Status_Change() == Sw ${}_{j}$  do        List P ${}_{i}$  =  all_k_shortest_path(Sw ${}_{i}$, Sw ${}_{j}$);        if  Sw ${}_{m}$   ∈ P ${}_{i}$  then               graph_list G ${}_{l}$  = graph.monitor.path - Sw ${}_{m}$;          Identify_attack( Sw ${}_{m}$,Port P ${}_{i}$);          call k_shortest_path(Sw ${}_{i}$, Sw ${}_{j}$) ;       else            return P ${}_{i}$;    end if    end ifend for     

To provide security and retain the resilience in the network, we propose DA-DIS as shown in Figure 1, which detects and isolates malicious switches. DA-DIS consists of two key modules named (1) detection system module (2) topology view of the update-module. The detection module mainly consists of two components named (1) ML classifier, and (2) decision module. It is possible that OpenFlow switches, which may causes delay, may not be malicious; therefore, it is necessary to consider all four scenarios which are mentioned in [8]. Therefore, the detection system module is responsible to detect the OpenFlow switch which delays LLDP packets maliciously. In the detection system, the ML classifier is responsible for making the decision based on the feature sets mentioned in the Figure 2. Statistical data for the classifier are provided by the monitor module from the switches of the data plane. Based on the provided statistics, the classifier makes its decision about the switches, whether it is malicious or normal. The topology view update module is responsible for updating the network topology as per the decision made by the detection system. If the decision system decides that the OpenFlow switch is malicious, then the topology view update-module will remove that particular switch from the path. Furthermore, it will notify this decision to the monitor module, which is responsible for notifying the resilience manager. The RM conducts the route-handoff in accordance with this decision and the malicious switch is isolated from the routing path. In this way, the DA-DIS system also achieves high resilience by isolating the malicious switch from the routing path.

## 4. Results and Analysis

To assess the effectiveness of the proposed mechanism, we evaluated two performance metrics: average network throughput and success rate (determined by the number of times the delay requirements of flows were satisfied).

### 4.1. Variable Number of Fault Events (20% Malicious Switches)

#### 4.1.1. Case 1: Link Break Event (E1)

In the process of the experiment, the connection (E1) between distinct switches in the network was disrupted one to five times. As shown in Figure 3, 20% of the network switches were under intrusion and normal settings. Under such circumstances, we compared the results of the success rate of the DA-DIS approach with SDN-RM under the delay attack. As the results depict, DA-DIS has a higher success rate. As DA-DIS can segregate malicious switches, it delivered a better success probability under the attack circumstances. If we compare DA-DIS with SDN-RM under attack, as well as in normal conditions, both give almost similar success rates, which means that DA-DIS can isolate the malicious switches. As the number of link failure events increased, there was a drop in the success rate in all three scenarios. As the frequency of connection failures rose, the success rates of all mechanisms reduced owing to the increasing number of faults. As a result, the success rate of DA-DIS declined when the network generated increased fault events. The average success rate was observed at 81.29% for DA-DIS in the case of the 20% (2 malicious switches out of 10) attack conditions in the 10-switch Mininet topology. On the other hand, the average success rates observed in SDN-RM under attack and conditions were 72.13% and 83.10%, respectively.

As shown in Figure 4, DA-DIS under the attack condition provided a higher network throughput as compared to SDN-RM under attack. It is evident that the network throughput dropped, as the magnitude of the connection failure rate. In the 20% attack scenario, the average throughput measured for DA-DIS was 954.2 Mbps while that of SDN-RM was 881.8 Mbps for one to five link failure events, while without an attack, SDN-RM provided a throughput of 971.4 Mbps.

#### 4.1.2. Case 2: Link Break Event and Dynamic Changes in Delay Requirements of Contract Events (E1 and E2)

Throughout the experiment, both fault events (E1 and E2) occurred one to five times at distinct time instances. As illustrated in Figure 5, DA-DIS under attack provided a higher success rate as compared to SDN-RM, while the success rate observed was almost similar for SDN-RM in a normal situation and DA-DIS in an attack situation. According to Figure 5, the success rate dropped as the number of events increased, with an average success rate of 80.75% for DA-DIS under the attack with 20 malicious switches. The success rate of SDN-RM under attack was 72.67%, which was lower compared to the DA-DIS. The average success rate for SDN-RM under normal circumstances was 80.95%.

As demonstrated in Figure 6, the network throughput deteriorated with an increase in the number of events (E1 and E2). DA-DIS provided higher throughput as compared to SDN-RM under attack. DA-DIS provided average network throughput of 967 Mbps for the E1 and E2 events under 20% malicious switches, while SDN-RM provided 883.4 Mbps throughput.

#### 4.1.3. Case 3: Dynamic Changes in Delay Requirements of Contract Event (E2)

In this experiment, the delay constraints (E2) were altered between one and five times at discrete intervals in the emulation, as depicted in Figure 7, e.g., the success rate and throughput, as well as Figure 8, decreasing with the number of events (E2) increasing. The success rates of DA-DIS and SDN-RM under attack were 80.27% and 69.33%, respectively. While DA-DIS and SDN-RM provided average throughput of 961.8 and 877.4 Mbps, respectively, under attack.

### 4.2. Variable Number of Malicious Switches

We increased the number of malicious switches in the 10 switches Mininet topology. We compared the success rate and throughput of DA-DIS with SDN-RM under attack. The number of malicious switches varied from 10% to 40% in the 10-switches network. Five fault events (E1 and E2) were investigated for the outcomes during the emulation. Other parameters remained the same as aforementioned.

#### 4.2.1. Case 4: Link Break Event (E1)

In this experiment, we progressively increased the total number of malicious switches in the network to compare the outcomes of DA-DIS and SDN-RM. In this experiment, the number of link failure events was induced twice at discrete time instances. Along with the link failure events, the percentage of malicious switches increased from 10% to 40% in the network. As shown in Figure 9, when the number of malicious switches increased in the network from 10% to 40%, the success rate decreased for SDN-RM. In contrast, DA-DIS provided a notable success rate as compared to SDN-RM with an increase in the percentage of malicious switches. The findings of Figure 10 demonstrate that the throughput of SDN-RM declined as the number of malicious switches rose, which resulted in a direct impact on the network’s QoS characteristics. While DA-DIS provided a notable throughput due to its capability to isolate malicious switches. Table 1 summarizes the results of Figure 9 and Figure 10.

#### 4.2.2. Case 5: Link Break Event and Dynamic Changes in Delay Requirements
of Contract Event (E1 and E2)

Both events (E1 and E2) occurred five times throughout the execution of the emulation, each time at a distinct time instance. The results of Figure 11 and Figure 12 are presented in Table 2 where the number of malicious switches increased from 10% to 40%.

#### 4.2.3. Case 6: Dynamic Changes in Delay Requirements of Contract Event
(E2)

Throughout the emulation, dynamic changes in delay requirements (E2) occurred five times at distinct time instances. The findings of Figure 13 and Figure 14 for this run are collected in Table 3.

### 4.3. Variable Number of Fault Events with Varying Number of Flows with
Real-World Test Bed

To evaluate DA-DIS, we carried out experiments in a real-world setup as shown in Figure 15, where three Zodiac Fx switches were connected with Port 4 to the system with the use of a TP-Link switch TL-SF1005D. The system ran the Ryu SDN controller on an Ubuntu virtual machine installed on a Windows 10 Enterprise (64-bit) host operating system. All three switches were connected with native ports forming a mesh topology. We took two computing systems as hosts, which were connected to Zodiac Fx switch-1 and Zodiac Fx switch-3, respectively. To access the terminals of both the hosts with *Putty* utility, USB to TTL serial cables were connected from the system to serial ports of both Raspberry Pis. To send TCP traffic from one host to the other, we used the iPerf tool. We considered sending each flow of 20 Mb with a time interval of 1 second. Table 4 summarizes the experimental parameters.

We varied the number of parallel flows (utilizing a higher percentage of bandwidth) from one to five for the source–destination pair (*Host 1* to *Host* 2) during this experiment. Moreover, we considered one malicious switch present in the SDN network.

#### 4.3.1. Case 7: Link Break Event (E1)

In this case, an event (E1) was triggered in order to generate a fault during the experiment. During the experiment, the link from switch 1 to switch 3 was broken once. As demonstrated in Figure 16, with the rising number of flows, the success rates of both approaches decreased due to increased network congestion [21]. As SDN-RM could not isolate the malicious switches from the routing path, it is evident that the average success rate of SDN-RM was 42.53% in the existence of a single malicious switch, while DA-DIS provided a higher success rate at 87.87%.

It is observed from Figure 17 that the average network throughput of SDN-RM was 39.22 Mbps, while that of DA-DIS was 53.59 Mbps in the presence of one malicious switch in the network.

#### 4.3.2. Case 8: Link Break Event and Dynamic Changes in Delay Requirements
of Contract Event (E1 and E2)

During the case, event E1 was triggered once and event E2 was triggered twice at distinct time instances. As shown in Figure 18, the success rate of DA-DIS was 76.27%, while for SDN-RM, it was 48.67%. It can be noted that, with the rising number of flows, the success rates of both approaches declined due to an increase in network congestion. It has been observed that from the linear fitting curves, DA-DIS had a higher success rate, which recites its ability to isolate the malicious switches from the routing path.

As demonstrated in Figure 19, DA-DIS and SDN-RM achieved average network throughput of 54.46 and 32.12 Mbps, respectively. Additionally, the linear fit curve demonstrates that DA-DIS offered greater throughput than SDN-RM.

#### 4.3.3. Case 9: Dynamic Changes in Delay Requirements of Contract Event (E2)

In this series of cases, we adjusted the number of parallel flows (using a larger percentage of bandwidth) for a source–destination pair from one to five by inducing various numbers of event E2s to produce a number of faults. In this case, we triggered the contract change event three times at separate time occurrences. As shown in Figure 20, the success rates of DA-DIS and SDN-RM were 80.53% and 48.53%, respectively. This was due to the aforementioned reasons.

As shown in Figure 21, DA-DIS provided average network throughput of 52.84 Mbps whereas SDN-RM provided 35.24 Mbps. As per the linear fitting of the curve, it is evident that the DA-DIS mechanism delivered superior throughput compared to SDN-RM even with a rising number of parallel flows. Thus, we can infer that DA-DIS has a strong capacity to achieve QoS parameters, which indicates a higher potential to achieve resilience in the presence of malicious switches in the network.

#### 4.3.4. Case 10: Effect on QoS Parameters of Malicious Switch during Emulation

In this case, we evaluated the proposed approach with an emulation time of 300 s in the presence of a malicious switch in the network. For this experiment, we utilized 60% of the bandwidth by generating 3 parallel numbers of flows. As shown in Figure 22, during the whole interval of emulation, the polynomial fit of the DA-DIS was higher than SDN-RM, which proves that DA-DIS has a higher success rate compared to SDN-RM. During this interval, none of them intersected with each other, meaning that DA-DIS has the capability of detecting and isolating the malicious switch in the network.

If we refer to Figure 23, the polynomial fit of the DA-DIS is higher than SDN-RM, which means DA-DIS provides high QoS performance in the presence of a malicious switch compared to SDN-RM.

*Observation:* It has been observed (from the above cases of experiments) that while the scaling of the attack increased, QoS parameters (success rate and average network throughput) were not affected. This exhibits the high resilience of the DA-DIS approach against the delay-based attack. However, an increasing parallel number of flow QoS parameters become affected due to the congestion.

## 5. Future Enhancement

In the future, we will aim to improve the resilience of the proposed scheme by mitigating distributed DoS (DDoS), warm-wholes, and deception attacks. Meanwhile, the current approach can be improved by considering communication with another device using an unknown protocol. Additionally, there is a scope for improving resilience by using multiple controllers in the case of a controller breakdown.

## 6. Conclusions

This work is an extension of the SDN-RM scheme, in which we intended to improve the resilience of SDN-RM under a delay-based attack. In this paper, we have presented a scheme for delay-based attack detection and isolation for time-critical applications. The approach detects the malicious OpenFlow switches in the SDN network that delays LLDP packets. An ML classifier detects the malicious switches in the network and the route-handoff mechanism isolates these switches from the routing path. As depicted by the emulation and real-world testbed results, the proposed scheme, DA-DIS, provides improved resilience with an increased average network throughput and success rate. DA-DIS shows higher resilience even with a higher percentage of attacker switches as compared to SDN-RM. Moreover, DA-DIS decreases the number of faults in the network, which is imperative for achieving zero-downtime in industrial networks.

## Figures and Tables

**Figure 1 sensors-22-06958-f001:**
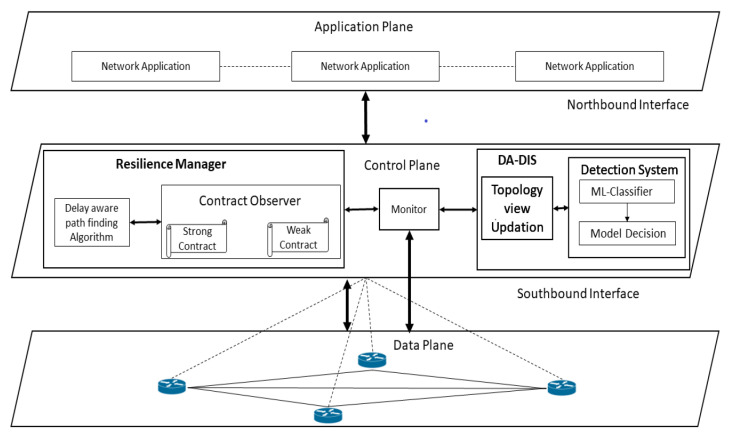
SDN framework for delay attack detection and isolation system (DA-DIS).

**Figure 2 sensors-22-06958-f002:**
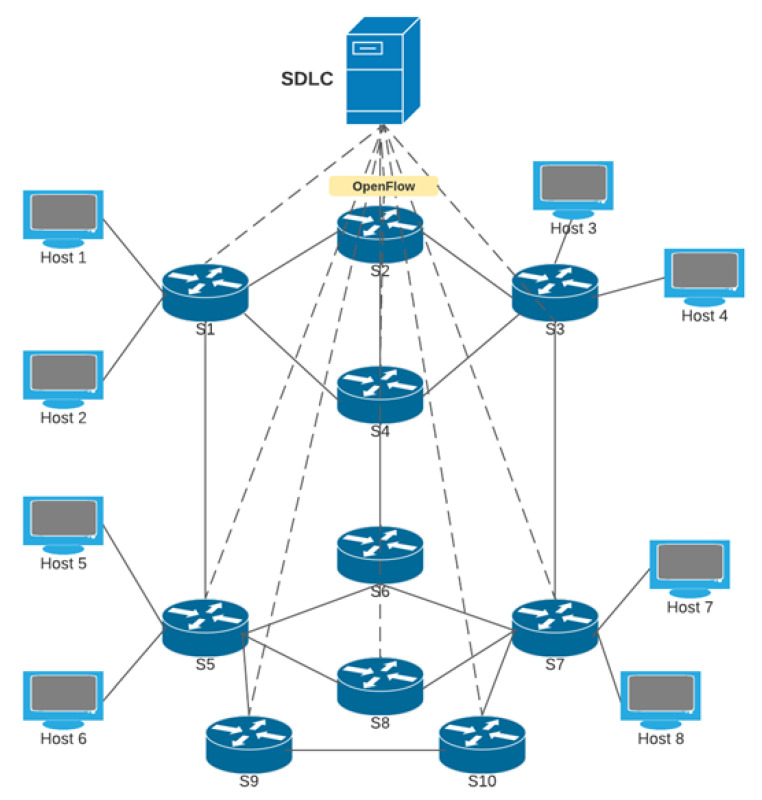
Mininet 10 switches emulation topology.

**Figure 3 sensors-22-06958-f003:**
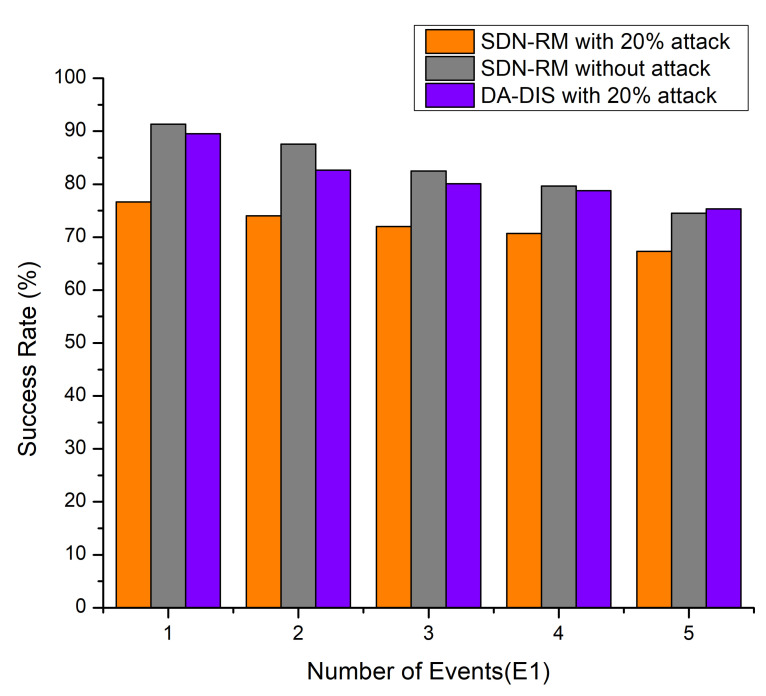
Success rate considering a different number of events (E1) and 20% malicious switches.

**Figure 4 sensors-22-06958-f004:**
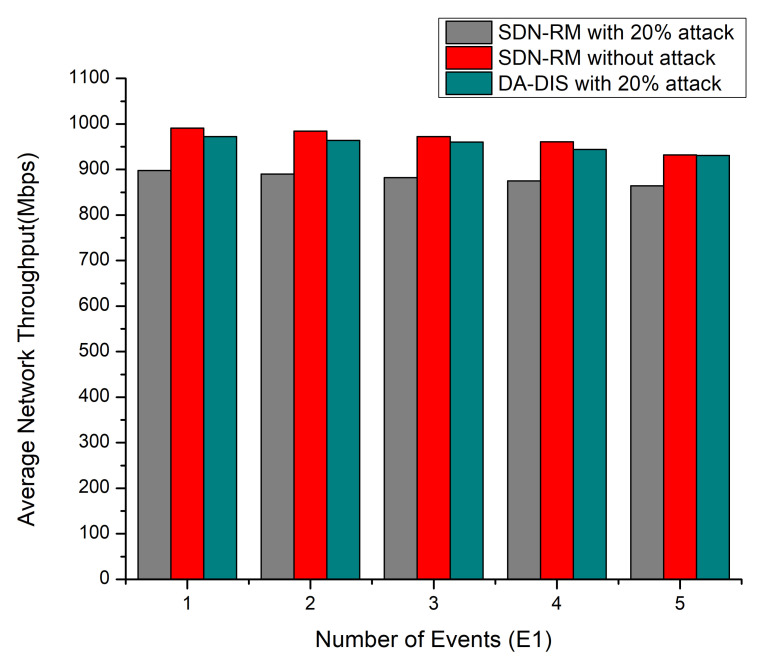
Throughput considering a different number of events (E1) and 20% malicious switches.

**Figure 5 sensors-22-06958-f005:**
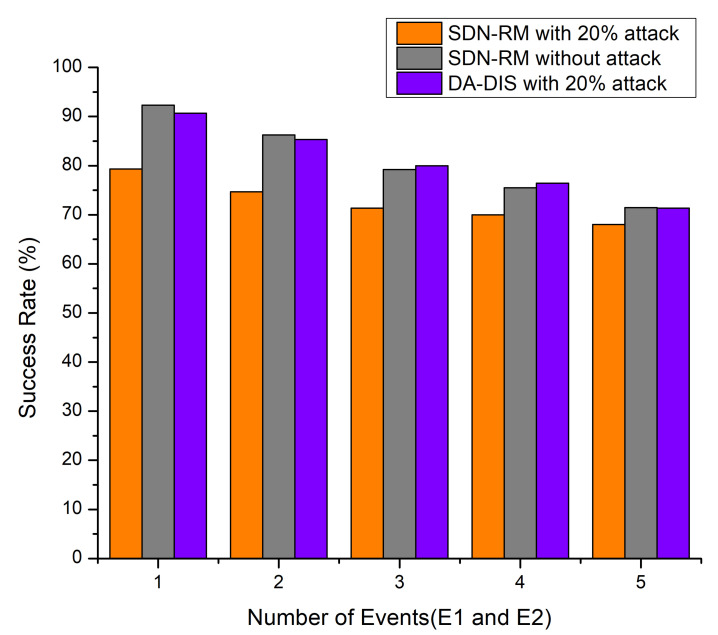
Success Rate considering a different number of events (E1 and E2) and 20% malicious switches.

**Figure 6 sensors-22-06958-f006:**
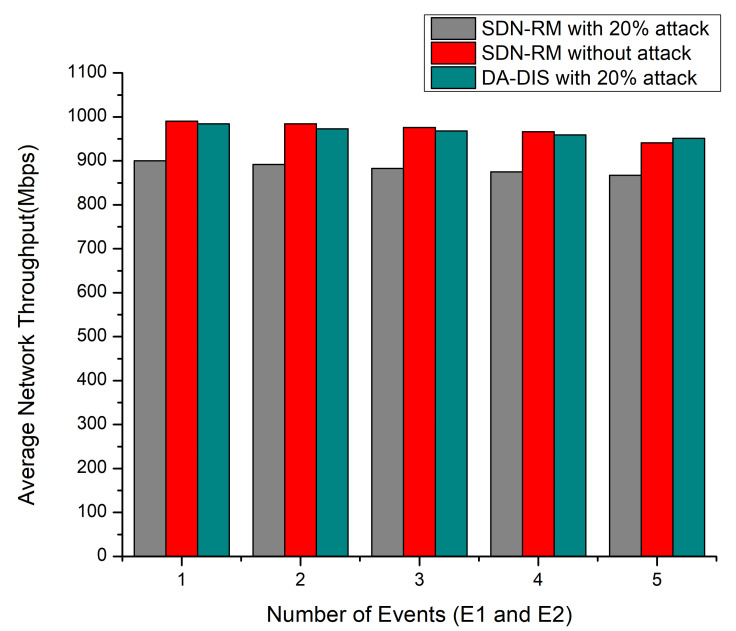
Throughput considering a different number of events (E1 and E2) and 20% malicious switches.

**Figure 7 sensors-22-06958-f007:**
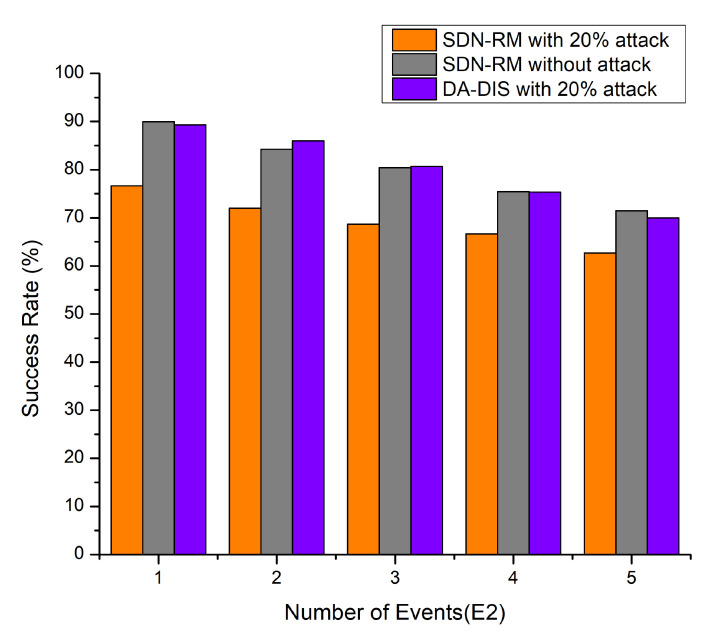
Success Rate considering a different number of events (E2) and 20% malicious switches.

**Figure 8 sensors-22-06958-f008:**
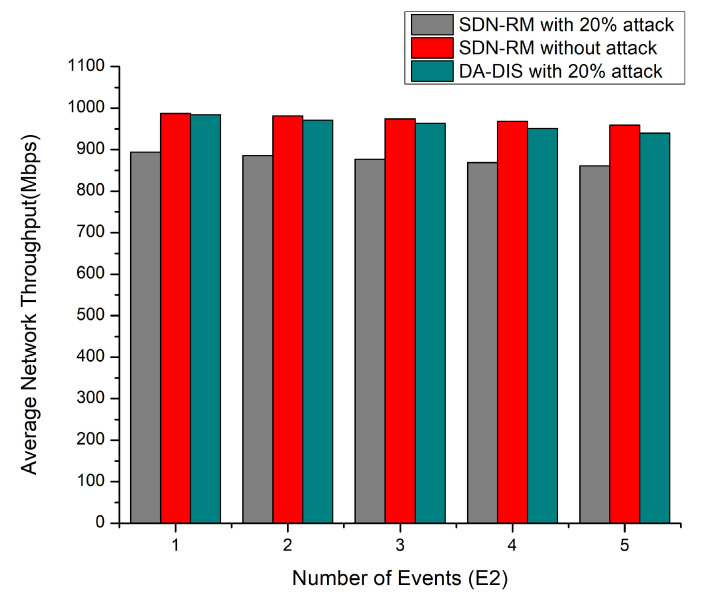
Throughput considering a different number of events (E2) and 20% malicious switches.

**Figure 9 sensors-22-06958-f009:**
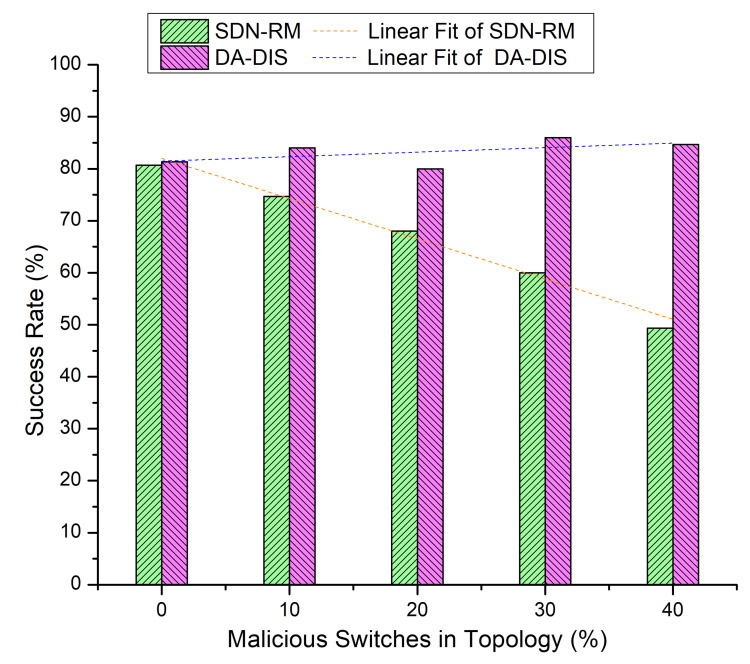
Success rate considering different numbers of malicious switches with E1 category faults.

**Figure 10 sensors-22-06958-f010:**
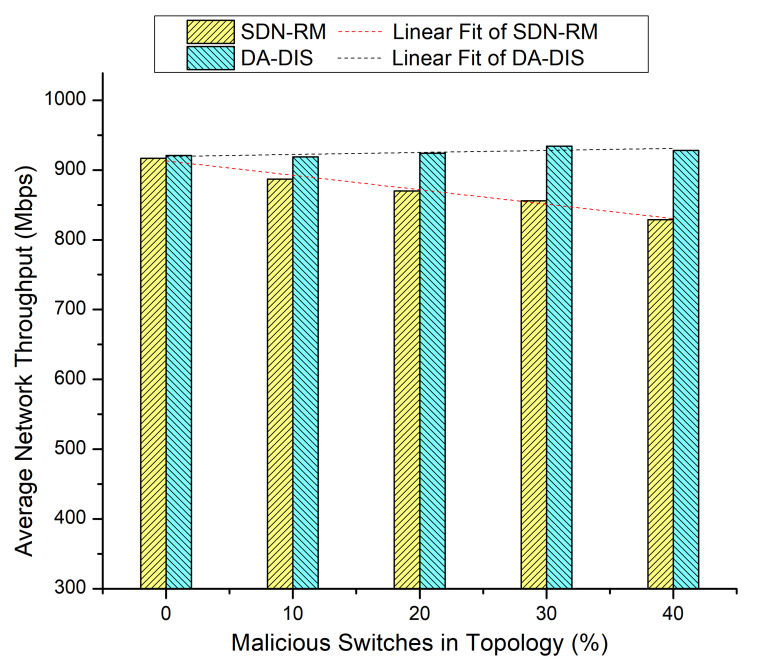
Throughput considering different numbers of malicious switches with E1 category faults.

**Figure 11 sensors-22-06958-f011:**
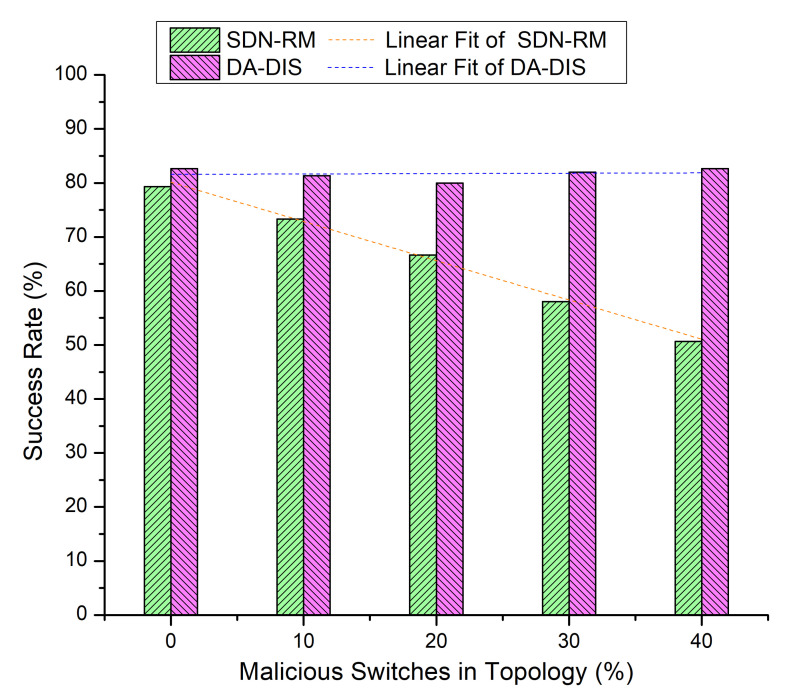
Success rate considering different numbers of malicious switches with E1 and E2 category faults.

**Figure 12 sensors-22-06958-f012:**
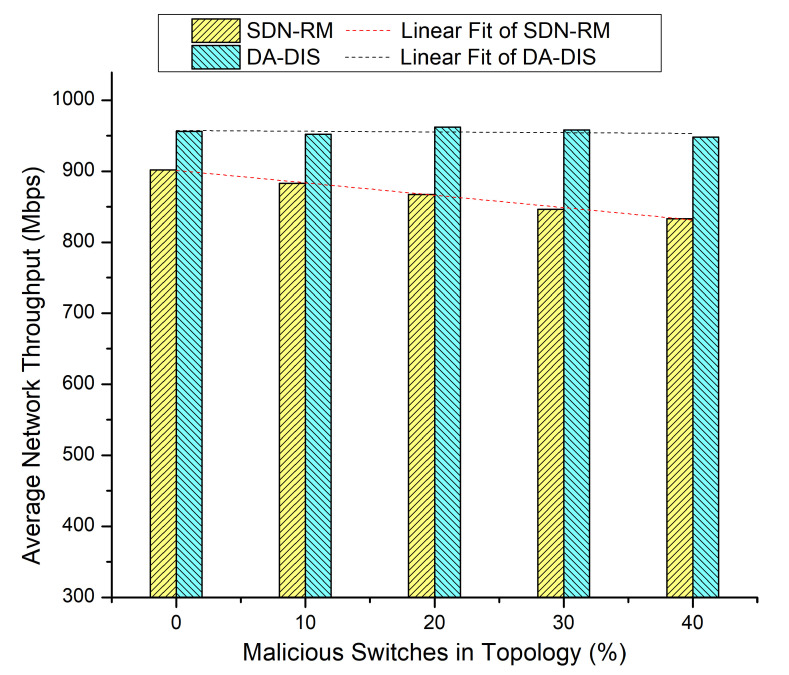
Throughput considering different numbers of malicious switches with E1 and E2 faults.

**Figure 13 sensors-22-06958-f013:**
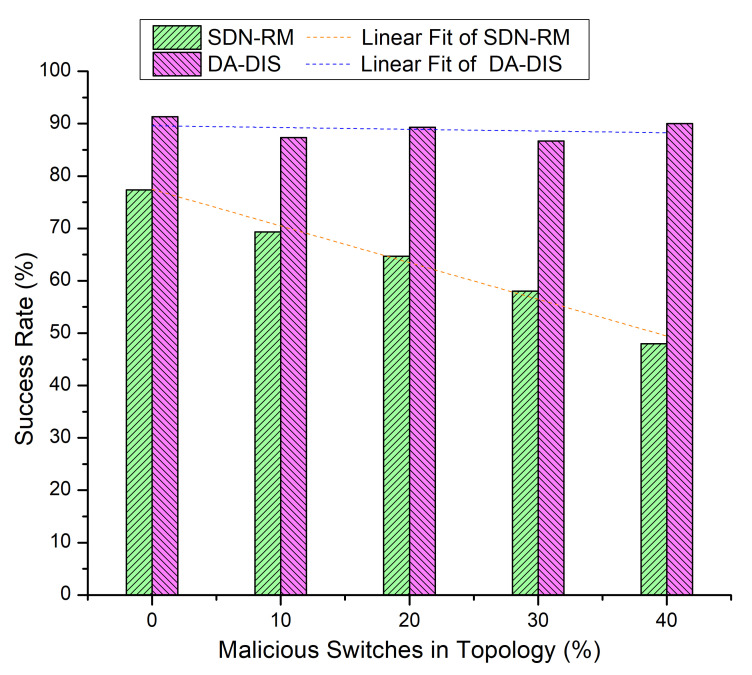
Success rate considering different numbers of malicious switches with E2 category faults.

**Figure 14 sensors-22-06958-f014:**
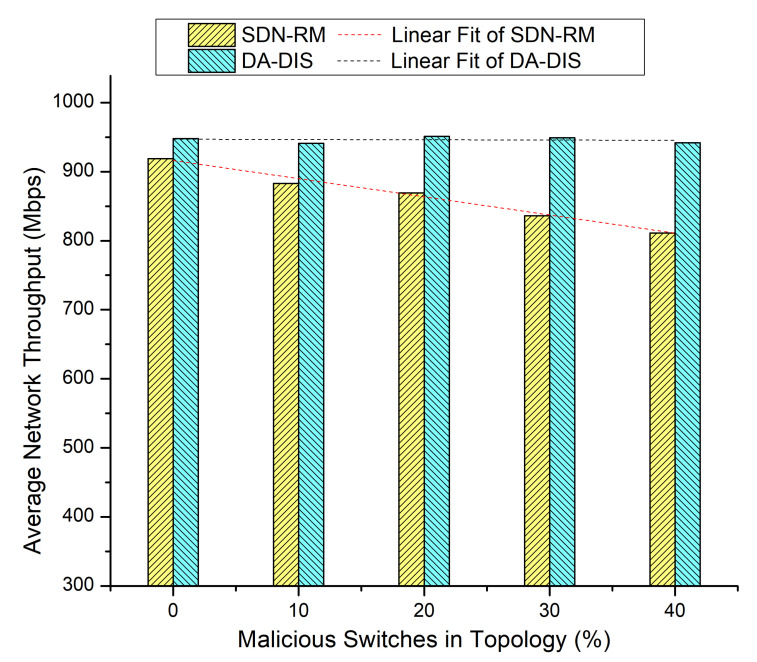
Throughput considering different numbers of malicious switches with E2 category faults.

**Figure 15 sensors-22-06958-f015:**
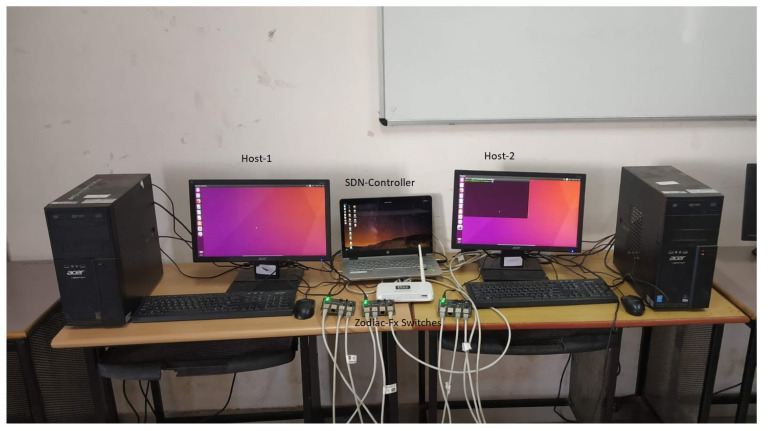
Real-world test bed.

**Figure 16 sensors-22-06958-f016:**
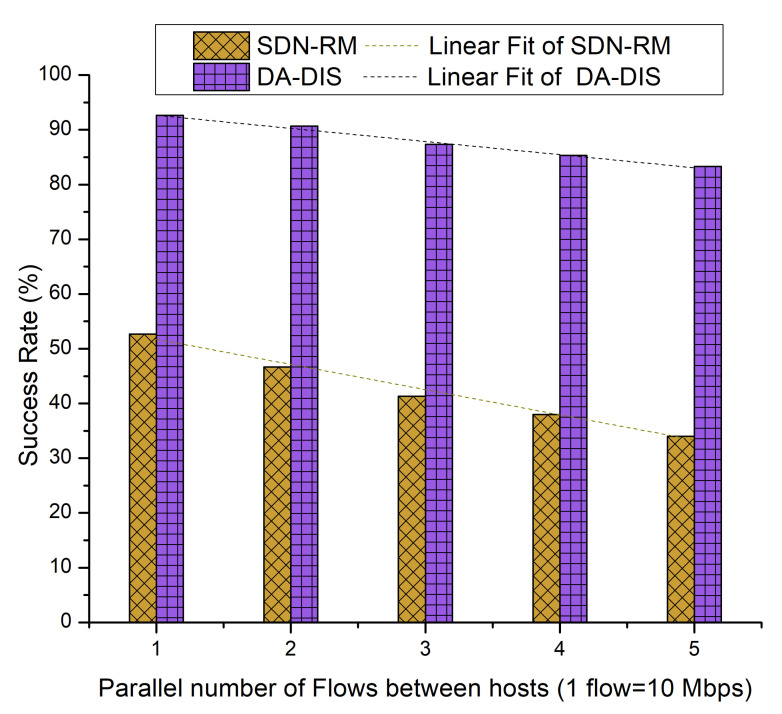
Success rate considering different numbers of parallel flows with one malicious switch along with E1 category faults.

**Figure 17 sensors-22-06958-f017:**
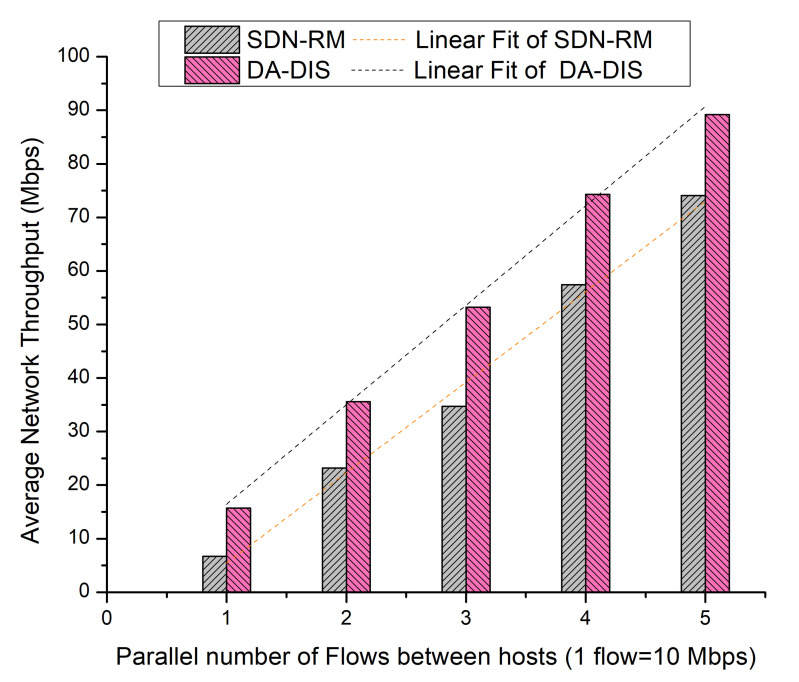
Throughput considering different numbers of parallel flows with one malicious switch along with E1 category faults.

**Figure 18 sensors-22-06958-f018:**
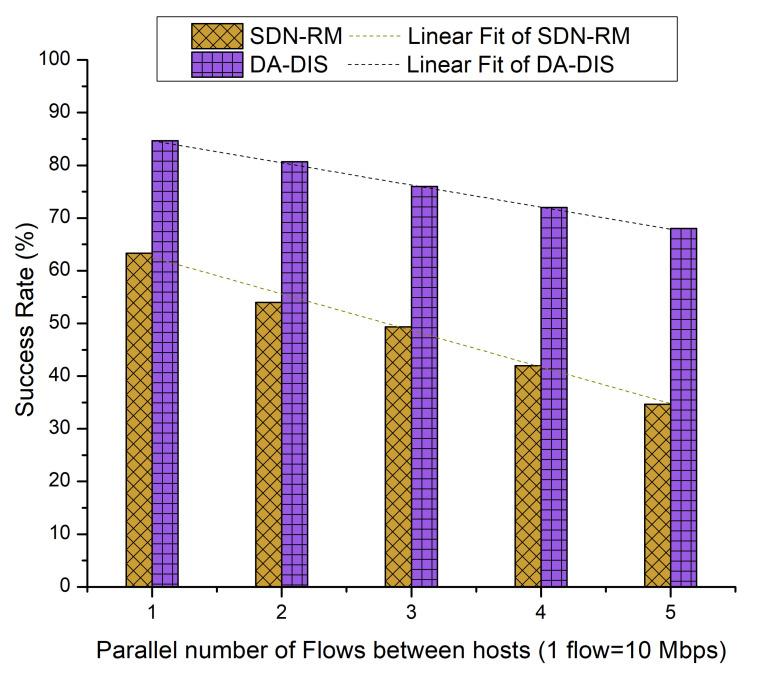
Success rate considering different numbers of parallel flows with one malicious switch along with E1 and E2 category faults.

**Figure 19 sensors-22-06958-f019:**
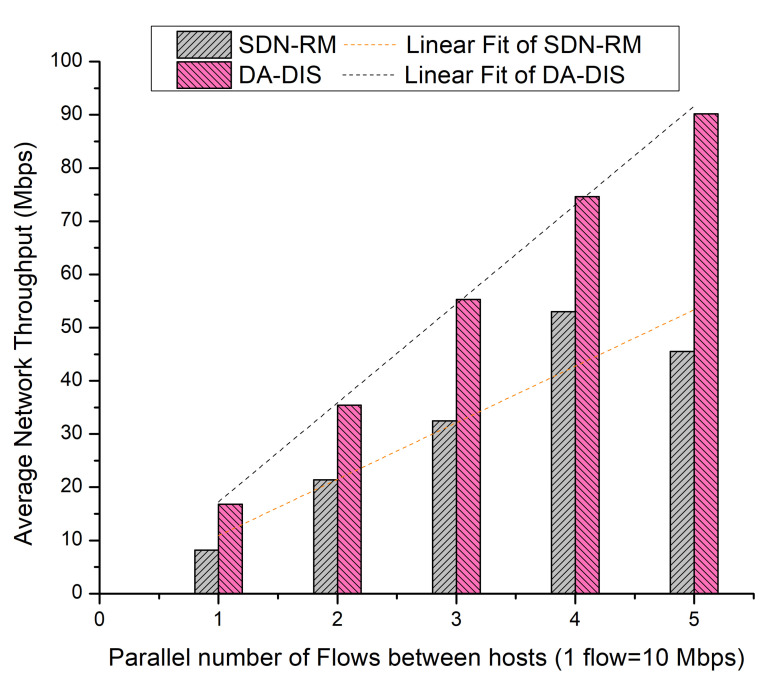
Throughput considering different numbers of parallel flows with one malicious switch along with E1 and E2 category faults.

**Figure 20 sensors-22-06958-f020:**
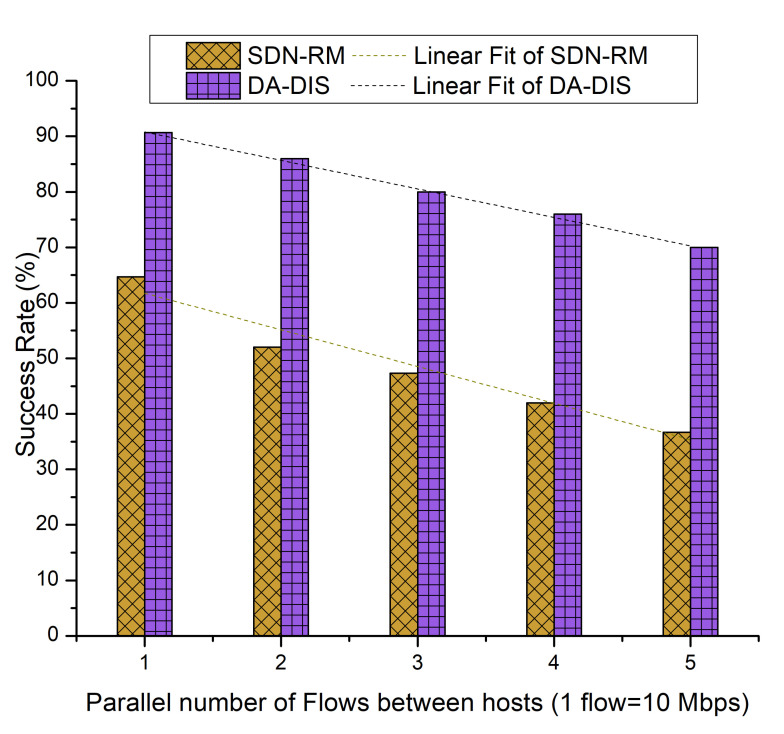
Success rate considering different numbers of parallel flows with one malicious switch along with E2 category faults.

**Figure 21 sensors-22-06958-f021:**
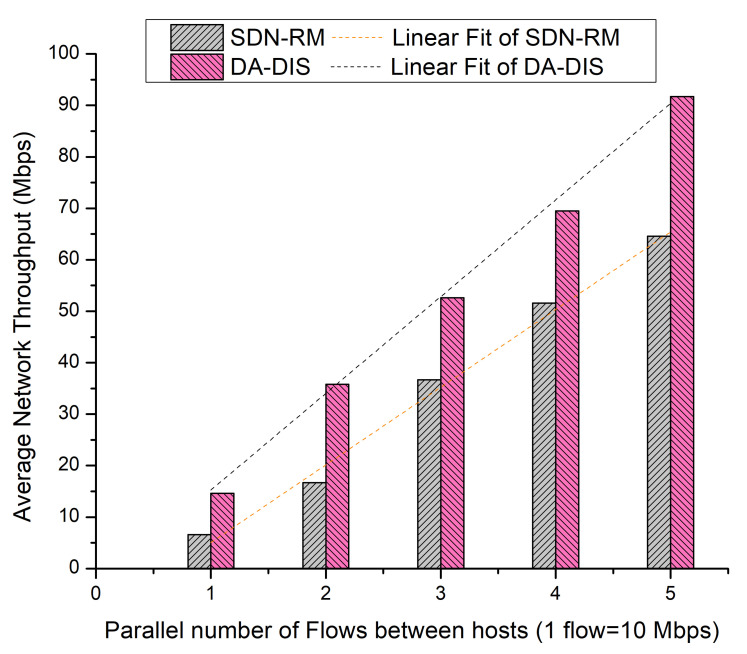
Throughput considering different numbers of parallel flows with one malicious switch along with E2 category faults.

**Figure 22 sensors-22-06958-f022:**
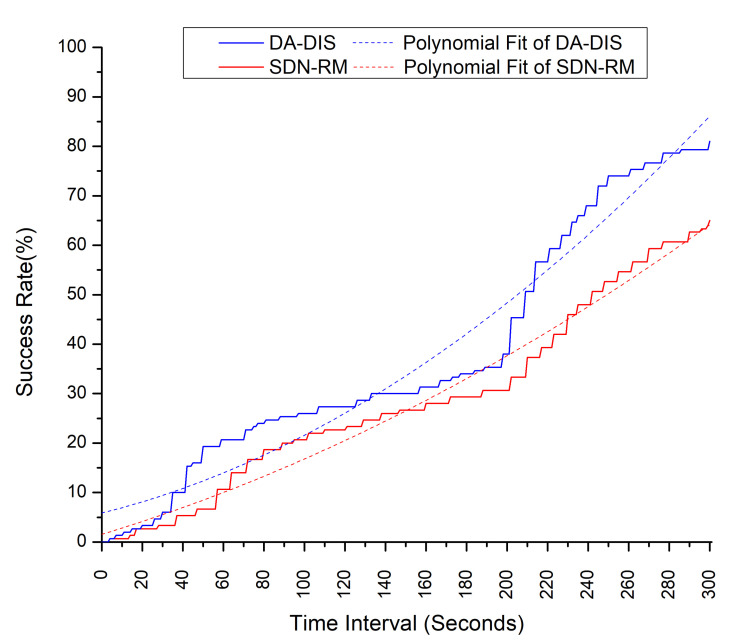
Effect of a malicious switch on the success rate during emulation.

**Figure 23 sensors-22-06958-f023:**
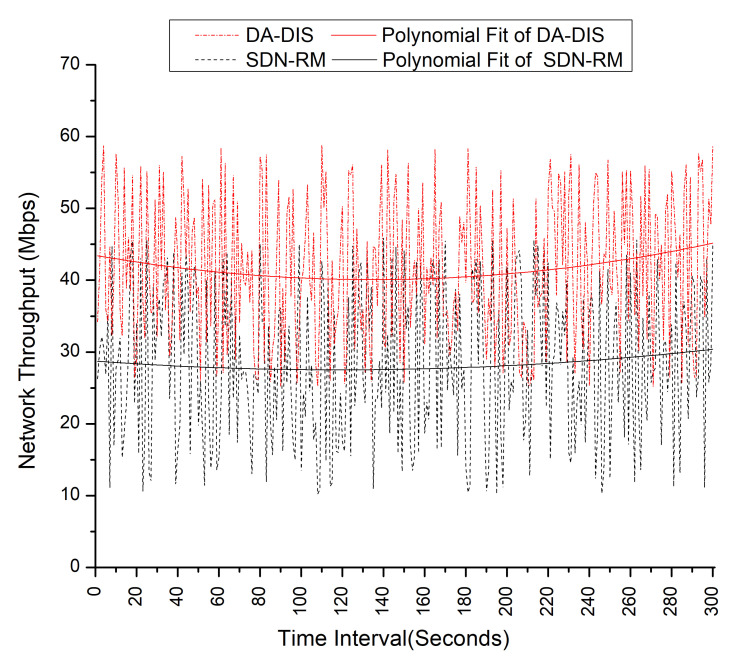
Effect of a malicious switch on throughput during emulation.

**Table 1 sensors-22-06958-t001:** Result overview of Case 4.

% of Malicious Switches in the Network	SDN-RM		DA-DIS	
	**Success Rate**	**Throughput**	**Success Rate**	**Throughput**
0%	80.67%	917 Mbps	81.33%	921 Mbps
10%	74.67%	887 Mbps	84.00%	919 Mbps
20%	68.00%	870 Mbps	80.00%	924 Mbps
30%	60.00%	856 Mbps	86.00%	934 Mbps
40%	49.33%	829 Mbps	84.67%	928 Mbps

**Table 2 sensors-22-06958-t002:** Results overview of Case 5.

% of Malicious Switches in the Network	SDN-RM		DA-DIS	
	**Success Rate**	**Throughput**	**Success Rate**	**Throughput**
0%	79.33%	902 Mbps	81.33%	956 Mbps
10%	73.33%	883 Mbps	84.00%	952 Mbps
20%	66.67%	867 Mbps	80.00%	962 Mbps
30%	58.00%	846 Mbps	86.00%	958 Mbps
40%	50.67%	833 Mbps	84.67%	948 Mbps

**Table 3 sensors-22-06958-t003:** Results overview of Case 6.

% of Malicious Switches in the Network	SDN-RM		DA-DIS	
	**Success Rate**	**Throughput**	**Success Rate**	**Throughput**
0%	77.33%	919 Mbps	91.33%	948 Mbps
10%	69.33%	883 Mbps	87.33%	941 Mbps
20%	64.67%	869 Mbps	89.33%	951 Mbps
30%	57.59%	836 Mbps	86.67%	949 Mbps
40%	48.00%	811 Mbps	90.00%	942 Mbps

**Table 4 sensors-22-06958-t004:** Experimental parameters.

Parameters	Value
Number of Zodiac Fx switches	3
Number of hosts	2
Number of flows (Varying)	1 to 5
Link capacity	100 Mbps
Time	300 s
Number of Events E1	1
Number of Events E2 (Varying)	1 to 3
Traffic type	TCP
Traffic generation	iPerf

## Data Availability

Not applicable.

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
