# Peer review of "ML-Based Delay Attack Detection and Isolation for Fault-Tolerant Software-Defined Industrial Networks"

_sensors, 2022, doi:10.3390/s22186958_

Round 1

Reviewer 1 Report

Dear Authors,

Revise the paper as per the following comments:

a)     How this research addressed the wormhole attacks and prevent them from manipulating Software-Defined Networks.

b)     In the introduction discuss the different types of mechanisms for securing the data pane.

c)     Multiple OpenFlow applications running over the controller might modify insertion and update the flow table. This led to a misconfiguration in the flow table how the authors addressed this issue in this research.

d)     Provide a table to analyze the literature review. What are the research gaps? How can you fill them? Compare the existing models in the literature and why you think your model is merit in the literature. Please work on improving the clarity of your paper.

There are many papers in this research area, and it is not clear whether the authors collected these papers based on which criteria. There is also some missing information in these papers. Please calibrate the reference section carefully.

Some closely related ones are recommended below:

10.3390/sym12050754

Authors may include them and comment on these previous works to highlight the motivation of this particular work.

e)     Is the proposed technique consumes less energy and provides less E2E delay? This needs some discussion.

f)      Algorithm 1, there is no input and output?

g)     Discuss SDN for industrial environments,

h)      Consider a network intrusion in which a port is suddenly disconnected, and another device is displayed thereon and begins communicating using an unknown protocol. How will this technique handle this issue?

i)      How the contributions were done. Please write the contributions clearly.

j)      The problem that has been solved in this research (state a hypothesis - a suggested solution to the problem).

k)     The datasets that have been used in the research experiments.

l)      Don’t cite any paper in the abstract.

m)   Add a new section named Research Gap before the conclusion section and discuss what the main limitations that you summarized are and what your main suggestions to solve these limitations are.

Author Response

Pl find the attached file

Reviewer 2 Report

This paper presents a real-time delay attack detection and isolation scheme for fault-tolerant software-defined  industrial networks, and a machine learning (ML)-based attack detection and isolation mechanism which extends our previous work,  SDN-RM is developed. Overall, the topic studied in this paper is practical. However, the following major comments are required to be considered.   

1. What does the abbreviation "SDN-RM [2]" mean in Abstract? In addition, usually, there is no citiation in Abstract. 

2. The challenage problems or difficulties of realizing the ML-based delay attack detection should be discussed and highlighted more clearly. 

3. Is the proposed ML-based attack detection strategy effective for other types of attacks, for example deception attacks?  

4. Recently, some meaningful results study the attack-tolerant problems from the point view of control method, especially for cyber-physical systems with deception attacks. Some relevant references are given as:  10.1109/TSG.2022.3178976. In general, the authors would do well to provide a stronger introduction.

5. Some future investigations about extending the proposed method to the deception attacks discussed in Comment 4 are suggested to be considered.

6. There are some grammar errors and typos, please check the presentation carefully. 

Author Response

Pl find the attached file

Reviewer 3 Report

Authors present a real-time delay attack detection and isolation scheme for fault-tolerant software-defined industrial networks and propose a machine learning (ML)-based attack detection and isolation mechanism which extends their previous work. 

The paper is very well organized. The introduction provides sufficient background and include relevant references. The design of the research is appropriate. The results are clearly presented and visualized through diagrams. 

It is not typical for authors to cite references in the Abstract and state what their contributions are in the Introduction. I recommend authors to describe the methods used in more details. Results are not compared with those of other similar studies.

Author Response

Pl find the attached file.

Reviewer 4 Report

This paper identifies real-time attacks on industrial fault-tolerant software networks and provides a ML based approach to detect malicious switches which are part of the attack and reroute traffic to increase network resilience and throughput.

Major Comments:

(1) The introduction is well-written but I think most readers will be left wondering what could be the prospective scale of the attack and how many networks of the type described (fault-tolerant software-defined industrial networks) exist currently in the world. The paper would be improved by setting context for the reader about the nature of the potential scale and impact. 

(2) The approach to identifying and explaining the attack seems related to the process described in by Padilla and Diallo in their paper on generating cyber attacks via monte carlo integration using markov chains. Discussing that the attack resembles properties of other well known attacks attacking in combination in a novel way - as identified in the reference - would help readers understand that this threat is significant and noteworthy.

Reference:

Gore, Jose Padilla, and Saikou Diallo. "Markov chain modeling of cyber threats." The Journal of Defense Modeling and Simulation 14.3 (2017): 233-244.

(3) A number of related references to the work presented in this paper are not included. The paper would be improved by a more thorough review of related work and differentiating the approach from these studies identified below:

Babiceanu, Radu F., and Remzi Seker. "Cyber resilience protection for industrial internet of things: A software-defined networking approach." Computers in industry 104 (2019): 47-58.

Das, Rohit Kumar, et al. "Ft-sdn: a fault-tolerant distributed architecture for software defined network." Wireless personal communications 114.2 (2020): 1045-1066.

Cheng, Chien-Fu, et al. "Reaching Consensus with Byzantine Faulty Controllers in Software-Defined Networks." Wireless Communications and Mobile Computing 2021 (2021).

Kreutz, Diego, Fernando MV Ramos, and Paulo Verissimo. "Towards secure and dependable software-defined networks." Proceedings of the second ACM SIGCOMM workshop on Hot topics in software defined networking. 2013.

(4) In the replication crisis error the data and source referenced in the paper, the scripts used for analysis, and the scripts used to create the figures for the paper need to be provided to both the reviewers and to the readership. This ensures completely transparent analysis and makes the authors paper significantly more impactful as other researchers can build off it.

(5) The paper makes numerous references to malicious switches (the term malicious is used 77 times in the paper). However, within the tables which present data the term "poisonous switch" is used without any introduction. If these terms are synonymous them only one of them should be used in the paper. If the terms mean something different then they need to be explicitly differentiated in the paper.

Minor comments:

(6) In the first column of table 3 and all columns of table 4 the data is center aligned which makes it difficult to read and compare rows to one another. For example in table three the 0% is not aligned with the ones digit of subsequent rows.

(7) The pseudocode in algorithm 1 makes reference to a number of functions that are not provided and not immediately intuitive to the reader. Providing the actual source code for this algorithm and referencing it in the paper at this point would improve things.

(8) There is a typo in at least one of the references. Kreutz et al.'s paper "Software-defined networking: a comprehensive survey" is spelled, "Software-dened networking: A comprehensive survey"

Author Response

Pl find the attached file.

Round 2

Reviewer 1 Report

Dear authors,

You have revised the paper very well. I could not see the recommended paper(which is close to your work) cited by you in the revised version. Kindly cite this paper 10.3390/sym12050754.

All the best!

Author Response

Response 1

Reviewer 2 Report

No further comments.

Author Response

Response 2

Reviewer 4 Report

The authors have addressed my concerns. The paper is now suitable for publication.

Author Response

Response 4
